# Surrogate-Assisted Pedestrian Protection Design via a Foundation Model–Orchestrated Workflow

**Osamu Ito**[*†]**, Akihiko Katagiri**[†]**, Yoshikazu Nakagawa,**
**Shin Saeki, Jun Shiraishi & Masato Sasaki**
Honda Motor Co., Ltd.
Wako-shi, Saitama 351-0188, Japan
`osamu_ito@jp.honda`

## Abstract

AI-driven engineering workflows face particular challenges in crash safety design: unlike aerodynamics, crash events involve highly nonlinear contact dynamics, material nonlinearity, and discrete state transitions that are difficult to capture with data-driven surrogate models. To the best of our knowledge, we present the first foundation model–orchestrated workflow for crash safety design that enables surrogate-assisted exploration for pedestrian protection, reducing evaluation time from hours per CAE simulation to seconds.

The workflow integrates four components: (1) a surrogate trained on CAE crash simulations to predict pedestrian leg injury metrics from design parameters, achieving an average $R^2 = 0.87$ and providing distribution-free conformal prediction intervals; (2) multiobjective evolutionary search (NSGA-II) to discover diverse feasible parameter sets under user-specified constraints; (3) a morphing-based geometry generator that maps parameters to topology-preserving 3D shapes; and (4) a natural-language interface in which an LLM orchestrates the workflow and a vision–language model supports semantic comparison of generated designs.

In an automotive front-bumper case study, the workflow produces 35 distinct safety-compliant alternatives from a single exploration, a process that would require weeks with conventional CAE iteration. These results suggest that foundation models can serve as integration layers between ML surrogates and physics-based simulation, helping bring AI capabilities to safety-critical engineering domains.

## 1 Introduction

Vehicle styling must satisfy aesthetic, safety, and packaging constraints simultaneously. In pedestrian protection, front-end geometry shapes how impact forces transmit to a pedestrian's leg, with injury outcomes depending nonlinearly on bumper, grille, and hood configurations. Conventional CAE simulations are high-fidelity but computationally expensive, limiting early-stage iteration. As in aerodynamics, when targets are not met, designers must revise the styling geometry and re-run CAE, creating an expensive trial-and-error loop.

Surrogate-based exploration has succeeded in aerodynamics, where performance depends largely on external surface geometry. Pedestrian protection exhibits the same iterative structure: failing injury targets implies modifying the front-end styling and re-evaluating. Crash safety is fundamentally different, however: injury outcomes depend on both external geometry and *internal structural response*—including deformation, energy absorption, and load transfer. This styling–structure coupling yields high-dimensional and potentially discontinuous responses that are challenging for data-driven modeling.

---

[*]Corresponding author. [†]Equal contribution.

Our key insight is that, for early-stage styling exploration, this coupling can be deliberately simplified without losing the relationships most relevant to initial design decisions. Specifically, we replace detailed structural models with parametric spring–mass representations, reducing the design space to 10 dimensions where surrogate modeling becomes tractable. Combined with morphing-based geometry generation and LLM-based orchestration, this enables surrogate-assisted exploration for crash safety design.

Our contributions are: (1) a novel foundation model–orchestrated workflow for crash safety design (Section 3); (2) a deliberate CAE data simplification strategy that makes surrogate modeling tractable for early-stage styling exploration (Section 4); and (3) an integrated design loop that combines diversity-preserving search, morphing-based geometry generation, and VLM-based semantic comparison (Sections 5–7).

## 2 RELATED WORK

**Surrogate modeling for automotive safety.** Surrogate models have been explored as computationally efficient alternatives to physics-based simulations in automotive safety design (Ito et al., 2019; Kaushik et al., 2021). Most prior studies emphasize predictive accuracy and use surrogates primarily as standalone evaluators, without embedding them into end-to-end design exploration pipelines that include constraint-aware search and geometry generation. Recent graph- and mesh-based approaches extend surrogate modeling to crashworthiness and explicit structural dynamics (Li et al., 2025a; Kneifl et al., 2025; Nabian et al., 2025), but their applicability is often limited to narrowly defined components or restricted geometric variations. In comparison, our work integrates surrogate prediction, evolutionary search, and geometry generation into a closed design loop orchestrated by foundation models.

**From aerodynamics to crash safety.** In aerodynamic design, surrogate models have progressed beyond standalone predictors and have been combined with design exploration and geometry generation, enabling surrogate-assisted, styling-oriented optimization of 3D shapes (Chen & Ramamurthy, 2021; Jabón et al., 2024; Vatani et al., 2025). Comparable end-to-end workflows have not been demonstrated for crash safety, largely due to differences in physics and data characteristics, including discontinuous contact interactions, strong material nonlinearity, and limited dataset availability resulting from time-intensive crash simulation setups.

**Design space exploration.** Multi-objective evolutionary algorithms such as NSGA-II (Deb et al., 2002) are widely used to explore trade-offs in engineering design, but their direct application to crash safety is often impractical due to expensive CAE-based fitness evaluation. Surrogate-assisted evaluation is a common strategy to mitigate this cost (Jones et al., 1998). In our setting, we exploit NSGA-II's population diversity mechanism not primarily for Pareto optimization, but for generating diverse feasible designs under safety constraints.

**Geometry generation and foundation models in engineering.** Recent advances in learned 3D generation (e.g., VAEs, diffusion models, NeRFs) have raised expectations for data-driven geometry synthesis. However, safety-critical engineering imposes stringent requirements: generated shapes often violate physical constraints, lack CAE-compatible topology, and cannot guarantee the dimensional accuracy required for manufacturing. For crash safety in particular, even small geometric deviations can induce large changes in injury outcomes due to contact nonlinearities.

We therefore adopt classical morphing-based generation (Sederberg & Parry, 1986; Samareh, 1999), which guarantees topology preservation and dimensional control. Our novelty lies not in proposing a new geometry generator, but in integrating proven morphing techniques into an LLM-orchestrated workflow that makes them accessible through natural language.

Large language models and vision–language models have been explored as interfaces for engineering analysis (Picard et al., 2023; Baker et al., 2025), but they typically play assistive roles rather than coordinating end-to-end workflows. In contrast, our work presents a *hybrid system* that uses foundation models as workflow orchestrators for crash safety, coordinating surrogate evaluation, evolutionary search, and geometry generation into a coherent design loop.

## 3 WORKFLOW OVERVIEW

A central question in AI-assisted engineering is how foundation models should interface with established computational tools. Two paradigms are commonly considered: (1) *replacing* domain tools with learned models, or (2) *orchestrating* existing tools through intelligent coordination. For crash safety, paradigm (1) remains challenging because learned models do not yet match the fidelity and reliability of physics-based simulation required for safety-critical decisions. We therefore adopt paradigm (2) and build a **hybrid system** in which foundation models act as orchestrators that coordinate proven engineering components—surrogates, optimizers, and geometry generators—under user guidance.

**LLM-based task orchestration.** Given a user request, the LLM determines which workflow component to invoke, manages inter-component data flow, and presents results. Using structured outputs, it extracts task intent and parameters from natural language and maintains context across turns, enabling iterative refinement of constraints. We support three task types:

- **Performance evaluation:** The user requests evaluation of a design. The LLM invokes the surrogate model and returns predicted injury metrics, highlighting any values exceeding thresholds.
- **Design generation:** When performance is unsatisfactory, the user requests design alternatives. The LLM conducts a structured dialogue to collect constraint bounds (e.g., "How much can the hood position change?"), then invokes evolutionary search and morphing-based generation.
- **Design analysis:** After generation, the user can query properties of the generated designs (e.g., "Which design has the smallest change from the original?"). The LLM analyzes parameter data and provides answers.

Figure 1 shows the system architecture. We use GPT-4o for its instruction-following and structured output capabilities; however, the workflow is not tied to any specific model. Any LLM with comparable abilities for task classification, parameter extraction, and multi-turn dialogue can serve as the orchestrator.

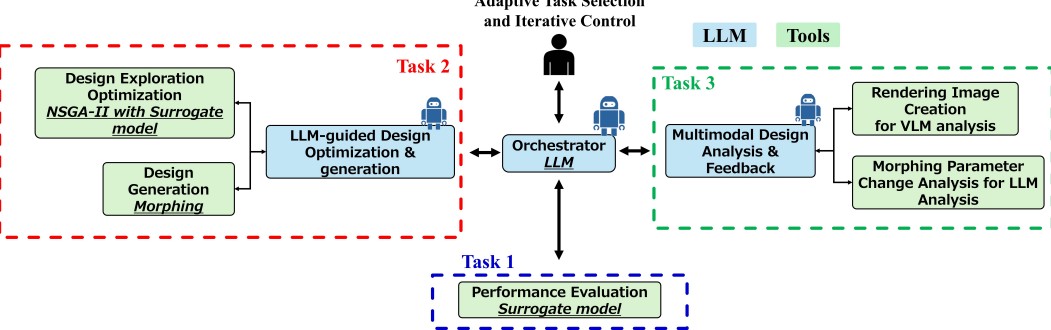

Figure 1: **LLM-based task orchestration.** The LLM interprets natural language, determines the task type, collects parameters through dialogue, and invokes modules. Users interact naturally while retaining full control over design constraints.

Figure 2 illustrates how orchestration executes in practice. Starting from a reference geometry, the workflow proceeds as follows:

- **Step 1:** The reference front-bumper geometry provides styling dimensions and load parameters.
- **Step 2 (Task 1):** The surrogate model predicts injury metrics for rapid performance evaluation.
- **Step 3 (Task 2):** If requirements are unmet, the LLM collects constraint bounds and invokes NSGA-II search (Section 5).

- **Step 4 (Task 2):** Feasible parameter sets are converted to 3D geometries via morphing (Section 6).
- **Step 5 (Task 3):** Generated designs are analyzed by the VLM for semantic comparison (Section 7).

Overall, the LLM selects the appropriate step based on user intent and supports iterative constraint refinement through multi-turn interaction.

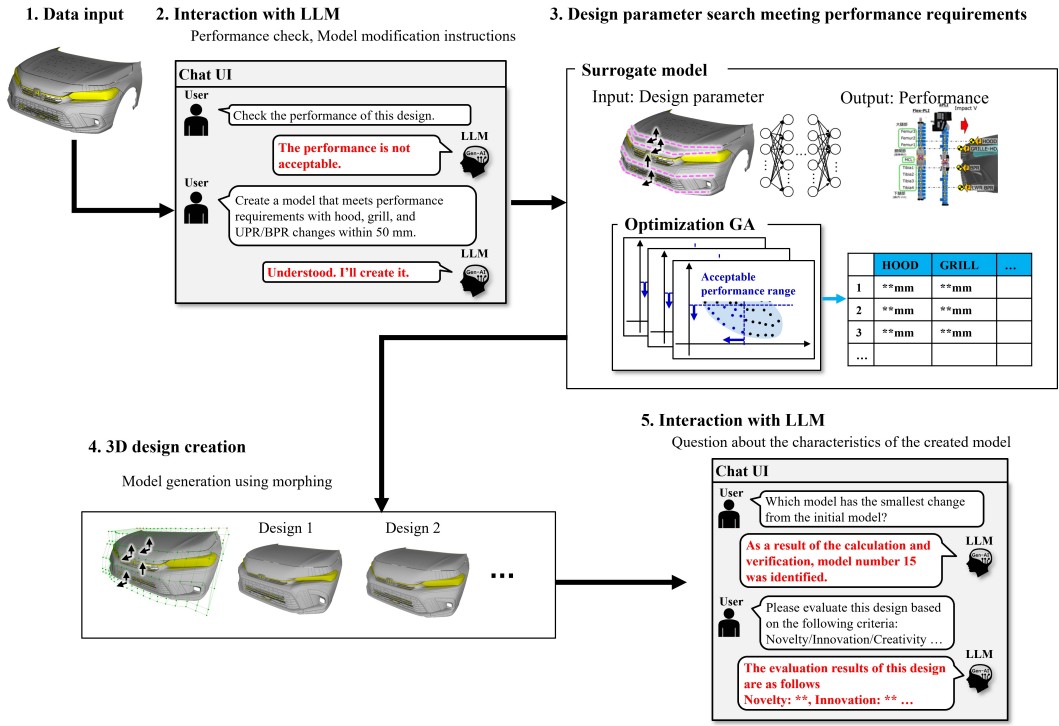

Figure 2: **Workflow execution.** Five steps from reference geometry to VLM-assisted comparison, with the LLM coordinating surrogate evaluation, evolutionary search, and morphing.

# 4 SURROGATE-GUIDED INJURY PREDICTION

## 4.1 PROBLEM FORMULATION

Pedestrian leg protection is assessed by impacting standardized leg-form impactors (FLEX-PLI and aPLI) against the vehicle front and measuring injury metrics. We consider thirteen indicators across both impactors: tibia bending moments (four locations) and knee ligament (MCL) elongation for both FLEX-PLI and aPLI, plus femur bending moments (three locations) for aPLI only. A design is feasible if all metrics remain below regulatory thresholds.

The injury response depends on both geometric styling parameters and structural load characteristics. We define a 10-dimensional design parameter vector $\mathbf{x} \in \mathbb{R}^{10}$ consisting of:

- **Geometric parameters** (7 variables): hood height and longitudinal position, grille height and longitudinal position, upper-bumper height, lower-bumper height and longitudinal position
- **Load parameters** (3 variables): reaction-force amplitudes for grille, upper bumper, and lower bumper (rectangular-wave loading)

The surrogate models learn mappings from design parameters to injury metrics: $f_{\text{FLEX}} : \mathbb{R}^{10} \to \mathbb{R}^{5}$ for FLEX-PLI (tibia moments and MCL) and $f_{\text{aPLI}} : \mathbb{R}^{10} \to \mathbb{R}^{8}$ for aPLI (tibia moments, MCL, and femur moments).

## 4.2 Training Data and Model Architecture

A central challenge in crash surrogate modeling is the tight coupling between external geometry and internal structural response. Detailed finite element models capture this coupling accurately but can induce high-dimensional and potentially discontinuous input–output relationships that are difficult to learn from practically obtainable datasets.

Rather than attempting to model detailed structural behavior, we focus on the requirements of early-stage styling exploration. At this stage, designers primarily need *relative* comparisons across geometric variations to guide the next styling update, while final verification can be deferred to high-fidelity CAE. This motivates a hybrid modeling approach:

- **Impactor side (high fidelity):** detailed finite element models of standardized leg-form impactors (FLEX-PLI, aPLI) preserve regulatory-compliant injury metric computation.
- **Vehicle side (intentionally simplified):** front-end structures are replaced with parametric spring–mass elements described by 10 parameters—7 geometric (hood, grille, bumper positions) and 3 load-related (reaction-force amplitudes).

This simplification transforms an intractable high-dimensional problem into a learnable 10-dimensional regression task. The trade-off is explicit: we sacrifice detailed structural accuracy to obtain a tractable surrogate, and selected candidates are subsequently validated with high-fidelity CAE. This is appropriate for early-stage exploration, where the goal is to identify promising styling directions rather than to finalize structural optimization. Accordingly, we employ standard ML regression instead of physics-informed approaches (e.g., PINNs, neural operators), because our workflow prioritizes capturing correlations sufficient for comparative evaluation and rapid screening.

Spring–mass parameters are linked to geometric modifications through morphing-based generation, enabling consistent updates of both shape and load characteristics. Training data are generated by randomly sampling 1,800 parameter combinations and running CAE simulations. Figure 3 illustrates the data generation process; Table 1 summarizes prediction accuracy (average $R^2 = 0.87$); and Appendix Figures A1 and A2 provide scatter plots comparing predictions against held-out test data for all 13 injury metrics.

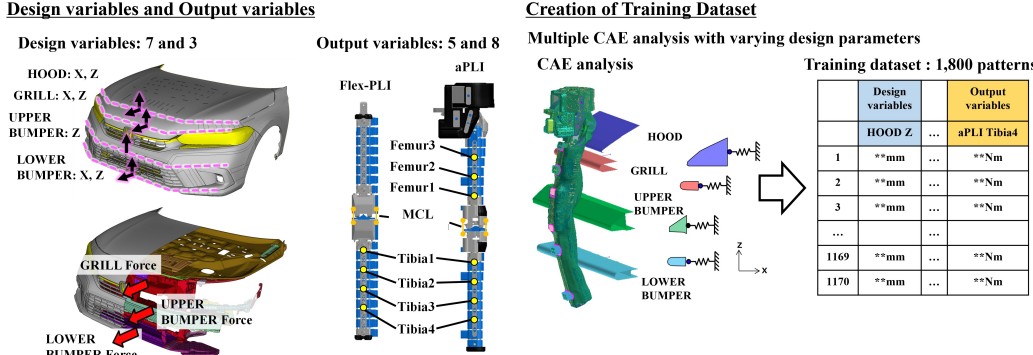

Figure 3: **Training data generation.** Left: 10 design parameters (7 geometric, 3 load). Hood reaction force is fixed because hood leading edge geometry is strongly constrained by styling requirements. Right: CAE simulation using a spring–mass vehicle model with detailed FE impactors (FLEX-PLI, aPLI). 1,800 parameter combinations were simulated.

## 4.3 Role of the Surrogate in the Design Loop

Within our workflow, the surrogate serves as a proxy for CAE, enabling rapid feasibility checks during search. Evaluating each candidate parameter set takes milliseconds rather than hours, which is essential for evolutionary optimization that assesses thousands of candidates per run. The surrogate preserves learned relationships between design parameters and injury outcomes; thus, identified

feasible regions correspond to safe designs, subject to subsequent high-fidelity CAE validation of selected candidates.

**Uncertainty quantification via conformal prediction.** Surrogate predictions inevitably carry errors. To quantify predictive uncertainty, we employ conformal prediction (Angelopoulos & Bates, 2021), which constructs prediction intervals without distributional assumptions under standard exchangeability conditions. Using a calibration set, we compute a quantile of absolute residuals required by conformal prediction and use it as the interval width, yielding distribution-free coverage guarantees for future predictions. This property is valuable for injury metrics computed via detailed finite element impactor models, where contact nonlinearities can produce non-Gaussian residuals. Table 1 reports 95% prediction intervals for each metric, helping designers identify candidates near safety thresholds that warrant additional CAE validation.

Table 1: **Surrogate model prediction accuracy and uncertainty quantification.** $R^2$ on held-out test data (20% of 1,800 samples). 95% PI (prediction interval) computed via conformal prediction provides distribution-free coverage guarantees. Units: Tibia and Femur in Nm; MCL in mm.

| Injury Metric | FLEX-PLI $R^2$ | aPLI $R^2$ | FLEX-PLI 95% PI | aPLI 95% PI |
|---|---|---|---|---|
| Tibia 1 | 0.81 | 0.81 | ±45.5 | ±49.1 |
| Tibia 2 | 0.90 | 0.89 | ±37.8 | ±42.5 |
| Tibia 3 | 0.94 | 0.87 | ±32.1 | ±41.4 |
| Tibia 4 | 0.95 | 0.80 | ±18.4 | ±25.3 |
| MCL (ligament) | 0.94 | 0.90 | ±3.1 | ±3.3 |
| Femur 1 | — | 0.84 | — | ±50.2 |
| Femur 2 | — | 0.80 | — | ±63.6 |
| Femur 3 | — | 0.80 | — | ±61.8 |
| **Average $R^2$** | **0.87** | | — | |

## 5 Constraint-Satisfying Design Space Exploration

### 5.1 Evolutionary Search Formulation

The goal of design exploration is not to find a single optimal solution, but to generate multiple feasible alternatives that satisfy safety requirements while offering diverse styling options. We formulate this as a constraint satisfaction problem: identify parameter combinations for which predicted injury metrics remain below safety thresholds.

Let $\mathbf{x} \in \mathbb{R}^{10}$ denote the design parameter vector. The search seeks parameter sets satisfying:

$$f_i(\mathbf{x}) \leq \tau_i \quad \forall i \in \{1, \ldots, 13\} \tag{1}$$

where $f_i(\mathbf{x})$ is the surrogate-predicted value for injury metric $i$ and $\tau_i$ is the corresponding safety threshold. Additionally, dimensional changes are bounded by user-specified constraints:

$$\mathbf{x}_{\min} \leq \mathbf{x} \leq \mathbf{x}_{\max} \tag{2}$$

where bounds are expressed relative to the reference design (e.g., hood height ±20 mm from baseline).

We employ NSGA-II for this search not for multi-objective optimization per se, but for its *population diversity preservation* mechanism. Standard single-objective optimizers tend to converge to a narrow subset of the feasible region, whereas NSGA-II's crowding distance helps maintain a spread of solutions. In our formulation, all thirteen injury metrics are treated as constraints rather than competing objectives; the algorithm is used to populate the feasible region with diverse parameter sets, rather than to trace a Pareto front. This yields multiple feasible designs corresponding to different styling options.

### 5.2 Implementation and Integration

We use a population size of 100 and run the search for 5 generations, evaluating 500 candidate designs per exploration. With surrogate evaluation taking milliseconds per candidate, the exploration completes in seconds, compared with weeks for direct CAE-based evaluation.

To ensure *practical* diversity, we discretize parameter values to 1 mm resolution, producing meaningfully distinct styling alternatives rather than near-identical clustered solutions (see Appendix Figure A6).

User-specified bounds are collected through the LLM interface. A typical interaction might specify: "hood position $\pm 20$ mm, grille position $\pm 50$ mm, upper bumper height $-10$ to $+100$ mm." The LLM parses these natural language constraints into the numerical bounds required by the optimizer. Feasible parameter sets identified by the search are passed to the geometry generation module.

## 6    STRUCTURE-PRESERVING 3D GENERATION

While recent 3D generative models show impressive visual results, they are often unsuitable for safety-critical engineering applications where *every millimeter matters*. Crash injury outcomes can be highly sensitive to geometric details, and uncontrolled variations introduced by generative models would undermine a surrogate-guided workflow.

We therefore adopt morphing-based generation—a deliberate choice that prioritizes reliability over novelty. Morphing provides: (i) precise dimensional control aligned with surrogate parameters, (ii) topology preservation to maintain CAE compatibility, and (iii) smooth interpolation that avoids introducing artificial discontinuities.

Morphing is controlled by a sparse set of control points embedded in a bounding box. As shown in Figure 4, control points are placed at 100 mm intervals and are tied to the same geometric styling parameters used by the surrogate (hood, grille, and upper/lower bumper positions). This shared parameterization ensures that designs deemed feasible by the surrogate correspond to realizable geometries. We implement morphing using an industry-standard CAE pre-processing tool.

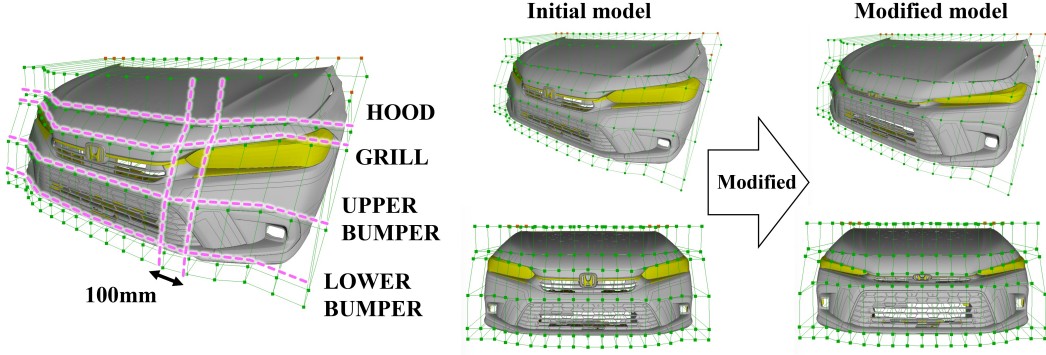

Figure 4: **Structure-preserving 3D generation via morphing.** Control points arranged at 100 mm intervals parameterize the front-end geometry and enable smooth, topology-preserving shape variations aligned with the surrogate's design parameters (hood, grille, and upper/lower bumper positions).

## 7    VLM-BASED SEMANTIC DESIGN COMPARISON

When exploration yields multiple feasible candidates, designers often need support to compare qualitative differences that are not captured by numerical objectives. We use a VLM to provide structured semantic comparisons that complement quantitative screening.

For each candidate, we generate four-view renderings (isometric, front, side, top) with a grid overlay for spatial grounding. Prior work has shown that axis-grid scaffolds—coordinate scales and grids overlaid on images—can improve VLM spatial grounding (Li et al., 2025b). Our 100 mm grid serves this role, enabling the VLM to quantify dimensional differences by counting grid squares rather than relying solely on qualitative descriptions. The VLM (Gemini 2.5 Pro) processes the images with prompts to:

- Describe salient visual characteristics
- Compare designs and identify specific differences
- Assess qualitative attributes (e.g., "sporty vs. mature", "aggressive vs. refined")

Figure 5 illustrates the VLM evaluation interface. This capability is intended to complement rather than replace human judgment: the VLM provides structured observations that help designers efficiently narrow down candidates, while final selection remains with human experts who can weigh factors the VLM cannot assess.

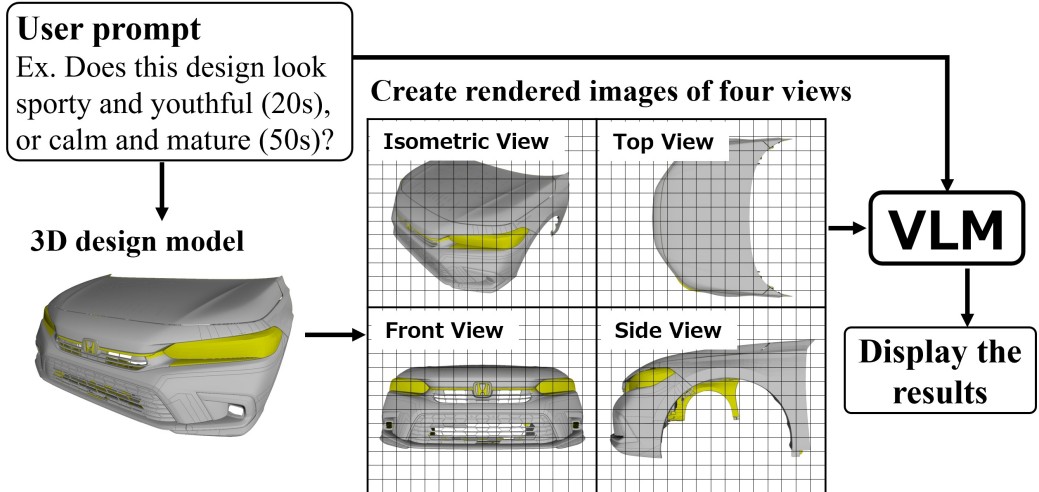

Figure 5: **VLM-based semantic evaluation of generated designs.** Multi-view renderings with grid overlays are processed by the VLM to provide qualitative descriptions and comparative assessments. The example shows evaluation of two design candidates on criteria including novelty, creativity, refinement, and appeal to different demographics.

## 8 CASE STUDY: FRONT BUMPER DESIGN

### 8.1 EXPERIMENTAL SETUP

We demonstrate the workflow using a production vehicle's front bumper as the reference design. To create a challenging test case, we set safety thresholds stricter than those used for the reference configuration, such that three injury metrics (aPLI Tibia2, aPLI Femur3, FLEX-PLI Tibia3) fail to meet requirements for the initial design. This setup emulates an early-stage design scenario in which styling modifications are needed to achieve safety compliance. For a detailed record of the human–LLM interaction, see Appendix Figure A3.

We explore the same 10-dimensional parameterization used by the surrogate. To reflect typical packaging constraints, allowable adjustment ranges are specified in millimeters through the LLM interface (see Appendix Table A1 for an example).

### 8.2 RESULTS: DESIGN GENERATION

The NSGA-II search (population 100, 5 generations) identified **35** feasible parameter sets that satisfy all injury constraints within the user-specified bounds.

Because surrogate evaluation takes milliseconds, the full search (500 candidates) completes in seconds, enabling interactive iteration on constraint bounds and rapid exploration of diverse styling alternatives.

Figure 6 highlights three representative feasible designs. **Design 1** shows minimal deviation from the reference and preserves the original character. **Design 2** shifts design lines upward with a forward-

| | Default | Design 1 (Minimum change) | Design 2 (Upward shift of the design line) | Design 3 (Sharp headlight) |
|---|---|---|---|---|
| **Isometric View** | | | | |
| **Side View** | | | | |

Figure 6: **Three representative designs from 35 feasible candidates.** Each satisfies all pedestrian leg injury constraints while exhibiting distinct styling: (1) minimal deviation from reference; (2) elevated design lines with prominent grille; (3) sharper headlight impression from differential upper/lower adjustments.

protruding grille, resulting in a more aggressive appearance. **Design 3** raises the upper bumper while lowering the grille, yielding a sharper headlight impression. Despite identical safety constraints, these examples exhibit distinct styling, indicating diverse aesthetic possibilities within the constrained design space.

### 8.3 RESULTS: VLM-BASED DESIGN EVALUATION

We evaluated whether the VLM can support semantic design comparison. In sanity checks, it identified salient visual features and distinguished headlight-related shape differences using the grid overlay.

We then asked the VLM to compare Designs 2 and 3 on five criteria (novelty, creativity, refinement, consistency, emotional appeal) and to assess which design would appeal more to users in their 20s. It characterized **Design 2** as more refined and broadly appealing and **Design 3** as more novel and creative, and judged **Design 2** more likely to appeal to users in their 20s.

These results suggest that VLM-based analysis can provide structured qualitative feedback that complements numerical feasibility metrics. Representative VLM evaluation examples are provided in Appendix Figs. A4 and A5.

## 9 CONCLUSION

To the best of our knowledge, we presented the first foundation model–driven workflow for crash safety design, integrating surrogate-based injury prediction, diversity-preserving design exploration, morphing-based geometry generation, and natural language/vision interfaces. Applied to pedestrian protection design, the workflow generated 35 diverse safety-compliant alternatives from a single exploration in seconds—a task that would require weeks using conventional CAE iteration.

In practice, this enables rapid early-stage styling iteration: designers express intent through explicit bounds, generate multiple safety-compliant alternatives in seconds, and use multimodal analysis to shortlist candidates for downstream CAE confirmation.

**Broader impact.** While surrogate-guided design exploration has flourished in aerodynamics, crash safety has remained difficult to access due to the complexity of impact physics. Our results suggest that foundation-model–orchestrated hybrid workflows may be useful for other "difficult" physical domains characterized by nonlinearities, discontinuities, and expensive simulations—settings where combining ML with physics-based tools can provide substantial practical value.

**Relevance to foundation models for science.** We position foundation models as workflow interfaces that orchestrate established engineering components under user-specified constraints, enabling natural interaction while remaining aligned with domain-standard evaluation criteria.

**Limitations and future work.** The surrogate (average $R^2 = 0.87$) requires high-fidelity CAE revalidation for selected designs. Conformal prediction provides distribution-free prediction intervals, but input-dependent uncertainty estimation remains future work. The 10-dimensional parameterization omits structural detail, and VLM–designer alignment has not been validated. Future directions include adaptive conformal methods for tighter intervals, active learning near feasibility boundaries, and richer surrogate models (e.g., neural operators) to expand applicability.

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

AUTHOR CONTRIBUTIONS

Osamu Ito (corresponding author) conceived the project, designed the workflow, implemented the surrogate modeling and optimization pipeline, and wrote the manuscript. Akihiko Katagiri, Yoshikazu Nakagawa, Shin Saeki, Jun Shiraishi, and Masato Sasaki contributed to discussions, provided domain expertise, and reviewed the manuscript.

# A ADDITIONAL FIGURES

Figure A1: **FLEX-PLI surrogate model prediction accuracy on held-out test data.** Scatter plots comparing CAE simulation results (x-axis) with surrogate predictions (y-axis) for 5 FLEX-PLI injury metrics (Tibia 1–4, MCL). The diagonal dashed line indicates perfect prediction. $R^2$ values range from 0.81 to 0.95, with an average of 0.91.

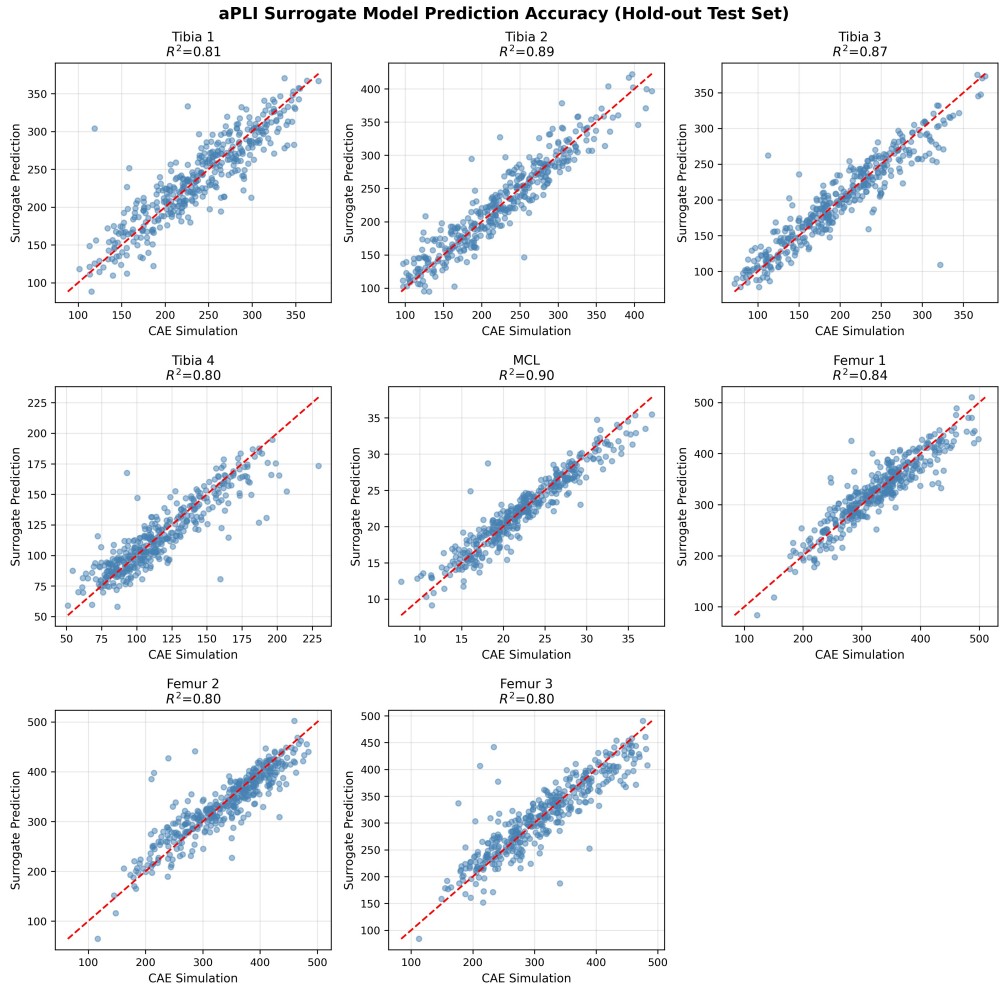

Figure A2: **aPLI surrogate model prediction accuracy on held-out test data.** Scatter plots comparing CAE simulation results (x-axis) with surrogate predictions (y-axis) for 8 aPLI injury metrics (Tibia 1–4, MCL, Femur 1–3). The diagonal dashed line indicates perfect prediction. $R^2$ values range from 0.80 to 0.90, with an average of 0.84.

| | Dialogue |
|---|---|
| User | Please evaluate the leg injury performance of this design. |
| System | The LLM determines that Task 1 should be executed. |
| LLM | After evaluating the leg injury performance, this design does not satisfy the required thresholds for aPLI Tibia2, Femur3, and FLEX-PLI Tibia3. Would you like me to assist with improving the design? |
| User | Please create a design that satisfies the performance requirements. |
| System | The LLM determines that Task 2 should be executed and initiates a clarification dialogue. |
| LLM | To generate a revised design, please provide the following geometric parameters and their allowable ranges: Hood height, Grille height, Upper bumper height, and Lower bumper height. |
| User | The hood height range should be $\pm 20$ mm. The grille height range should be $\pm 50$ mm. The upper bumper height should be between $-10$ mm and $100$ mm. The lower bumper height range should be $\pm 10$ mm. The horizontal adjustment range should be $\pm 20$ mm. |
| LLM | Based on your input, the design constraints are defined and confirmed. I have generated 35 candidate designs that satisfy the specified conditions. |
| User | Which design exhibits the smallest deviation from the initial design? |
| System | The LLM determines that Task 3 should be executed. |
| LLM | Among the generated candidates, the 18th design shows the smallest overall change from the initial design. The change is computed as the sum of absolute differences over the *geometric* parameters only (excluding load parameters), and the resulting score is 45.5, which is the minimum among all candidates. |

Figure A3: **Example human–LLM dialogue during surrogate-guided design exploration.** The conversation illustrates how the LLM interprets user intent and orchestrates the design workflow, including performance evaluation, constraint clarification, design generation, and selection of a candidate with minimal deviation from the initial design.

| Question | Input image | VLM's response |
|---|---|---|
| What is this image of, and which company makes this product? |  | This image appears to show a design drawing of a car's front bumper, which is the part located at the lower front of the vehicle. Since there is an emblem in the center, it is likely a product of an OEM. |
| Please compare the images of Model 1 and Model 2, and determine which one has the thinner light design? |  | Based on the grid as a reference, it can be determined that Model 2 has a thinner light design. Reason:

Model 1: The thickest part of the headlight (in the vertical direction) appears to be approximately the width of one grid square (about 100 mm).

Model 2: The vertical width of the headlight is clearly narrower than one grid square and appears to be about half the size (approximately 50 mm).

Therefore, it can be concluded that the headlight of Model 2 is vertically thinner in design compared to that of Model 1. |

Figure A4: **Representative VLM evaluation examples.** The table shows example interactions with the VLM, demonstrating its ability to identify visual features and compare design characteristics. The first row shows that the VLM identifies the object as a vehicle front bumper based on an emblem on the grille. The second row demonstrates grid-based spatial reasoning: when asked to compare headlight designs between two models, the VLM uses the grid overlay to estimate that Model 1 has a headlight width of approximately 100 mm (one grid square), while Model 2 has a narrower headlight of approximately 50 mm, correctly concluding that Model 2 has the thinner light design.

| Question | VLM's response |
|---|---|
| Please compare Design Proposals 2 and 3 from the provided images based on the following five criteria, using a 5-point scale: novelty, creativity, sophistication, consistency, and emotional appeal. Additionally, please consider which design is more likely to be favored by people in their 20s. | **Design Proposal – Evaluation Summary**

**1. Evaluation Scores (out of 5):**
**Design Proposal 2**
Novelty: 3 Creativity: 3 Sophistication: 5 Consistency: 4 Emotional Appeal: 4 **Total: 19 points**
**Reason:** The design is highly refined and polished, giving a sharp and sporty impression with universal appeal.

**Design Proposal 3**
Novelty: 4 Creativity: 4 Sophistication: 3 Consistency: 4 Emotional Appeal: 3 **Total: 18 points**
**Reason:** This design is rich in novelty and creativity, with a unique and bold character. However, it appears slightly complex and is less sophisticated compared to Proposal 2.

**2. Support from People in Their 20s:**
**Design Proposal 2 is more likely to be favored by people in their 20s.**
**Reason:** The sophisticated and sporty design aligns well with the preferences of today's younger generation. |

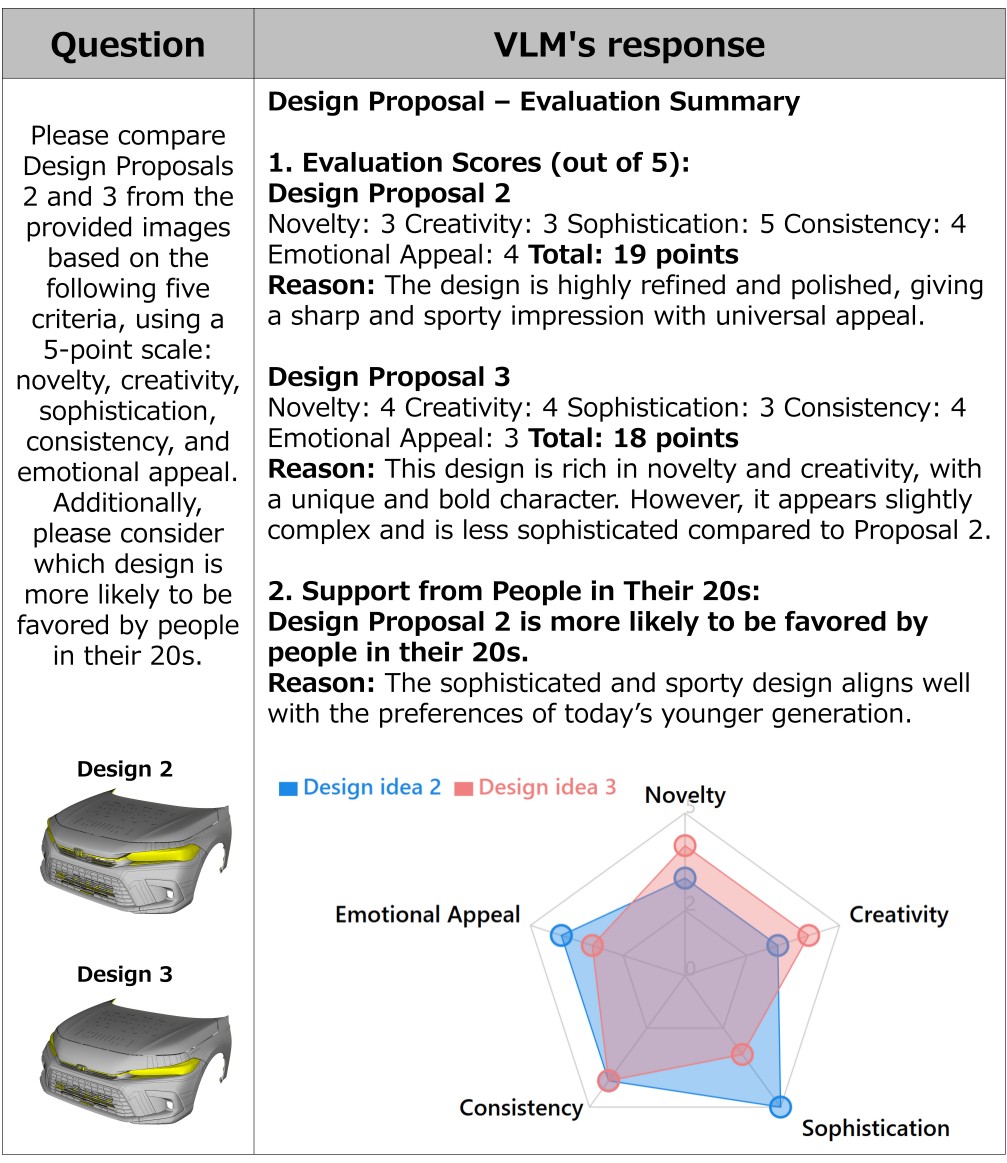

Figure A5: **VLM-based qualitative design assessment.** Multi-view renderings of generated designs are processed by the VLM, which provides structured qualitative assessments including novelty, creativity, refinement, and demographic appeal. These evaluations complement numerical feasibility metrics by capturing styling characteristics that are difficult to quantify.

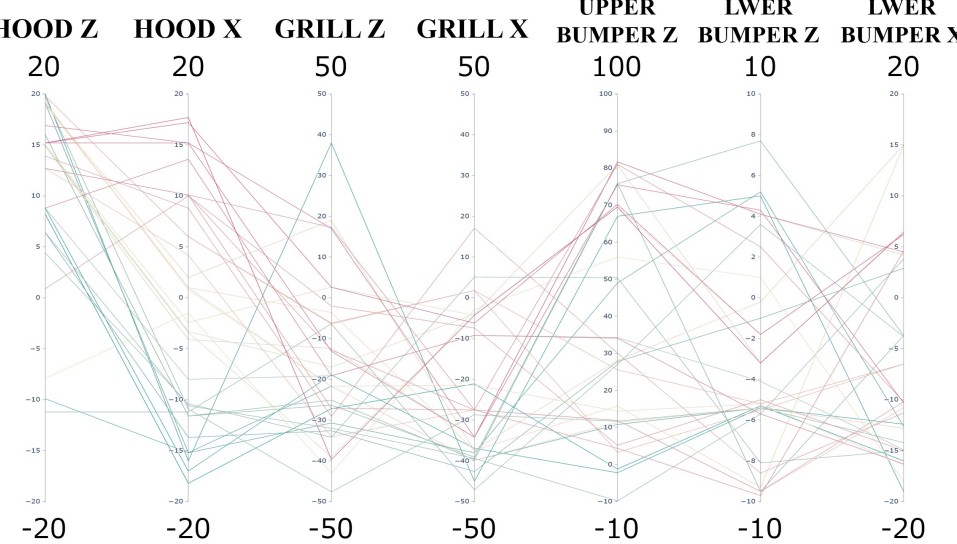

| HOOD Z | HOOD X | GRILL Z | GRILL X | UPPER BUMPER Z | LWER BUMPER Z | LWER BUMPER X |
|---|---|---|---|---|---|---|
| 20 | 20 | 50 | 50 | 100 | 10 | 20 |
| -20 | -20 | -50 | -50 | -10 | -10 | -20 |

Figure A6: **Diversity of generated feasible designs in parameter space.** Multi-dimensional visualization showing the distribution of 35 feasible designs across the 10-dimensional parameter space. Each axis represents a design parameter (hood, grille, and bumper positions), and each point corresponds to a feasible design identified by the NSGA-II search. The spread of solutions demonstrates that the discretization strategy (1 mm resolution) successfully produces diverse styling alternatives rather than clustered near-identical solutions.

Table A1: **Example constraint bounds specified through natural language.** Bounds are given relative to the reference geometry and reflect typical packaging constraints.

| Parameter | Min (mm) | Max (mm) |
|---|---|---|
| Hood height | $-20$ | $+20$ |
| Hood longitudinal | $-20$ | $+20$ |
| Grille height | $-50$ | $+50$ |
| Grille longitudinal | $-50$ | $+50$ |
| Upper bumper height | $-10$ | $+100$ |
| Lower bumper height | $-10$ | $+10$ |
| Lower bumper longitudinal | $-20$ | $+20$ |

## B  IMPLEMENTATION DETAILS

### B.1  SURROGATE MODEL SELECTION

For each injury metric, we performed exhaustive model selection across 14 regression algorithms: MLP Regressor, Gaussian Process Regressor (with various kernels), Gradient Boosting Regressor, Extra Trees Regressor, Random Forest Regressor, Bagging Regressor, K-Neighbors Regressor, AdaBoost Regressor, SGD Regressor, Theil-Sen Regressor, Huber Regressor, Passive Aggressive Regressor, Decision Tree Regressor, and RANSAC Regressor. The best-performing model was selected independently for each of the 13 injury metrics based on $R^2$ on held-out validation data. This per-metric selection allows different injury responses (which may have different underlying relationships with design parameters) to be captured by the most appropriate model architecture.

### B.2  NSGA-II CONFIGURATION

Multi-objective search uses the following parameters: population size of 100, crossover probability of 0.9 (simulated binary crossover with $\eta_c = 20$), and mutation probability of $1/n$, where $n$ is the

number of variables (polynomial mutation with $\eta_m = 20$). The algorithm runs for 5 generations, evaluating 500 candidates in total.

## B.3 SOFTWARE AND MODELS

- Surrogate training: scikit-learn
- Multi-objective optimization: pymoo
- Morphing: ANSA (BETA CAE Systems)
- LLM: GPT-4o (gpt-4o-2024-08-06)
- VLM: Gemini 2.5 Pro (gemini-2.5-pro-exp-03-25)

