# OpenReview forum: "Surrogate-Assisted Pedestrian Protection Design via a Foundation Model–Orchestrated Workflow"
_ICLR.cc/2026/Workshop/FM4Science — ICLR 2026 Workshop FM4Science Poster_

### Official Review · Reviewer_6w9f · 2026-02-19
**Solid Engineering Workflow with Thin Foundation Model Contribution**

**Rating:** 6
**Confidence:** 3

**Review:**

This paper presents an end-to-end workflow for early-stage pedestrian protection design, combining four components: (1) a surrogate model trained on 1,800 CAE crash simulations (avg $R^2 = 0.87$) with conformal prediction intervals, (2) NSGA-II search for diverse feasible designs under safety constraints, (3) morphing-based topology-preserving geometry generation, and (4) an LLM/VLM interface (GPT-4o + Gemini 2.5 Pro) for natural-language orchestration and semantic design comparison. A front-bumper case study yields 35 safety-compliant styling alternatives in seconds.

**Pros:**

- **Well-motivated and practically grounded.** The deliberate simplification of vehicle-side structure to a 10D spring–mass parameterization is clearly scoped to early-stage styling exploration, and the trade-off vs. full FE fidelity is honestly stated. This is thoughtful engineering for a real and difficult problem domain (crash nonlinearity, discontinuous contacts).
- **Complete pipeline with appropriate UQ.** Conformal prediction is a sensible choice for distribution-free coverage guarantees on injury metrics with potentially non-Gaussian residuals. The paper correctly positions conformal intervals as flags for designs near safety thresholds that warrant CAE re-validation.
- **Excellent writing and presentation.** Figures are clear, the workflow architecture is easy to follow, the LLM dialogue examples (Appendix A3) give concrete evidence of how the system works in practice, and the limitations section is refreshingly candid.

**Cons:**

- **The "foundation model" contribution is thin relative to the workshop scope.** GPT-4o performs task classification, parameter extraction, and multi-turn dialogue—essentially a chatbot UI layer. Gemini 2.5 Pro provides subjective aesthetic assessments ("sporty vs. mature"). Neither model encodes physical priors, symmetries, or scientific constraints—the qualities FM4Science specifically targets. The actual scientific modeling is done by standard scikit-learn regressors (14 off-the-shelf algorithms, best picked per metric). Removing the LLM/VLM and replacing it with a conventional GUI would not change the engineering substance.
- **No CAE validation of surrogate-selected designs.** For a safety-critical application, this is a significant omission. The paper generates 35 feasible candidates but never closes the loop by verifying even a subset against high-fidelity simulation. The workflow's practical value is premised on surrogate accuracy, yet the reader has no evidence that the feasible set actually corresponds to safe designs. The wide conformal intervals (e.g., ±50–64 Nm for femur metrics) sharpen this concern—are these intervals small or large relative to the safety thresholds? This is never discussed.
- **Limited novelty in individual components.** NSGA-II, FFD morphing (Sederberg & Parry 1986), conformal prediction, and LLM-as-interface are all well-established. The claimed contribution is integration, but the integration is relatively straightforward sequential tool-chaining (surrogate → optimizer → morphing → VLM rendering). There is no feedback loop where downstream results inform upstream choices, no active learning to refine the surrogate near decision boundaries, and no learned component that improves with use.
- **VLM evaluation is unvalidated.** The VLM's aesthetic judgments (novelty, creativity, appeal to specific demographics) are presented without any comparison to human designer assessments. The authors acknowledge this, but for a workshop audience interested in trustworthy AI for science, unvalidated VLM opinions on subjective design attributes feel premature as a contribution.
- **Tiny search budget masks diversity questions.** Only 5 generations with population 100 (500 total evaluations) were run. Whether the diversity of the 35 feasible designs is due to NSGA-II's crowding distance or simply to the small budget leaving the population unconverged is unclear. An ablation comparing random sampling or Latin hypercube sampling under the same budget would strengthen the claim.

**Overall:** A clean, well-written systems paper that demonstrates a practical engineering workflow for a challenging domain. The surrogate + conformal prediction + morphing pipeline is the strongest contribution. However, the foundation model component amounts to a convenience interface rather than a scientific modeling advance, and the lack of CAE validation for any generated design weakens the practical claims. For FM4Science specifically, the paper would benefit from either (a) incorporating physics-aware surrogate architectures, or (b) demonstrating that the LLM orchestrator provides capabilities beyond what a standard parametric interface could achieve. Marginally above the acceptance threshold given the practical relevance and honest scoping, but the workshop fit is a concern.

---

### Official Review · Reviewer_aCbQ · 2026-02-23
**This paper proposes an end-to-end, foundation model–orchestrated workflow for early-stage pedestrian protection design in automotive crash safety. By deliberately simplifying vehicle-side structural modeling and integrating surrogate prediction, diversity-preserving evolutionary search, morphing-based geometry generation, and LLM/VLM-based interfaces, the authors demonstrate that safety-compliant design alternatives can be generated in seconds rather than weeks**

**Rating:** 6
**Confidence:** 4

**Review:**

## Strengths

1. Clear motivation and problem framing
The paper convincingly explains why pedestrian crash safety is fundamentally more challenging than aerodynamics for surrogate-assisted design, grounding the work in real engineering constraints.

2. Pragmatic and well-justified simplification strategy
The deliberate reduction of structural complexity via a spring–mass vehicle model is explicitly tied to early-stage design needs. The trade-off between fidelity and tractability is clearly stated and appropriate.

3. Strong system-level integration
The paper goes beyond standalone surrogate modeling by presenting a closed-loop workflow that integrates prediction, search, geometry generation, and human interaction.


## Weaknesses

1. Limited novelty at the algorithmic level
Most individual components—regression-based surrogates, NSGA-II, morphing, and conformal prediction—are well established. The primary novelty lies in system integration rather than new methods.

2. Insufficient baseline comparisons
The surrogate model is evaluated mainly via R^2 and prediction intervals, without comparison to alternative surrogate formulations (e.g., joint multi-output models, physics-informed models, or neural operators). This weakens the empirical justification for the chosen design.

3. Dataset size and diversity are limited
The surrogate is trained on 1,800 simulations derived from a single simplified configuration. The relatively small and homogeneous dataset weakens the solidity and generalizability of the conclusions.

---

### Meta-Review · Area_Chair_W4gB · 2026-02-27

**Recommendation:** Accept (Poster)
**Confidence:** 4

**Metareview:**

The average review score is above 6, which means reviewers recommended an acceptance.

---

### Decision · Program_Chairs · 2026-03-03

Accept (Poster)